# Effects of Different Nanoparticles on Microbes

**DOI:** 10.3390/microorganisms11030542

**Published:** 2023-02-21

**Authors:** Bin Niu, Gengxin Zhang

**Affiliations:** 1State Key Laboratory of Tibetan Plateau Earth System, Resources and Environment (TPESRE), Institute of Tibetan Plateau Research, Chinese Academy of Sciences, Beijing 100101, China; 2University of Chinese Academy of Sciences, Beijing 100101, China

**Keywords:** nanoparticles, microbes, interactions, Fe (III), aggregation

## Abstract

Nanoparticles widely exist in nature and may be formed through inorganic or organic pathways, exhibiting unique physical and chemical properties different from those of bulk materials. However, little is known about the potential consequences of nanomaterials on microbes in natural environments. Herein, we investigated the interactions between microbes and nanoparticles by performing experiments on the inhibition effects of gold, ludox and laponite nanoparticles on *Escherichia coli* in liquid Luria–Bertani (LB) medium at different nanoparticle concentrations. These nanoparticles were shown to be effective bactericides. Scanning electron microscopy (SEM) images revealed the distinct aggregation of cells and nanoparticles. Transmission electron microscopy (TEM) images showed considerable cell membrane disruption due to nanoparticle accumulation on the cell surfaces, resulting in cell death. We hypothesized that this nanoparticle accumulation on the cell surfaces not only disrupted the cell membranes but also physically blocked the microbes from accessing nutrients. An iron-reducing bacterium, *Shewanella putrefaciens*, was tested for its ability to reduce the Fe (III) in solid ferrihydrite (HFO) or aqueous ferric citrate in the presence of laponite nanoparticles. It was found that the laponite nanoparticles inhibited the reduction of the Fe (III) in solid ferrihydrite. Moreover, direct contact between the cells and solid Fe (III) coated with the laponite nanoparticles was physically blocked, as confirmed by SEM images and particle size measurements. However, the laponite particles had an insignificant effect on the extent of aqueous Fe (III) bioreduction but slightly enhanced the rate of bioreduction of the Fe (III) in aqueous ferric citrate. The slightly increased rate of bioreduction by laponite nanoparticles may be due to the removal of inhibitory Fe (II) from the cell surface by its sorption onto the laponite nanoparticle surface. This result indicates that the scavenging of toxic heavy metals, such as Fe (II), by nanoparticles may be beneficial for microbes in the environment. On the other hand, microbial cells are also capable of detoxifying nanoparticles by coagulating nanoparticles with extracellular polymeric substances or by changing nanoparticle morphologies. Hence, the interactions between microbes and nanoparticles in natural environments should receive more attention.

## 1. Introduction

Nanosized earth materials are ubiquitous in nature, not only as individually definable particles but also in the form of other nanoscale domains and as features within and on minerals [1,2,3]. Nanoparticles are almost everywhere in the environment and exist in solid, liquid or gas phases [4,5,6,7]. In weathering processes, silicate, oxides, sulfide and other minerals are unstable under earth surface conditions and dissolve or react with water, oxygen and other reagents to form other mineral phases, such as clay, oxides, oxyhydroxides and hydroxides. These mineral phases are a few to a few tens of nanometres in size [8,9]. On Earth, both organic and inorganic nanomaterials are related to pure geosystems as well as biogeosystems [2,10].

In the environment, nanoparticles are formed via inorganic and organic pathways that usually involve high supersaturation degrees, leading to the generation of many crystal nuclei in geochemical processes. In organic pathways, supersaturation can occur as a result of fluctuations in the pH and redox state of ions that lead to changes in ion and mineral solubility. For example, iron (manganese) oxides and sulphide nanoparticles can be produced through iron reduction or oxidation by metal-oxidizing/reducing bacteria or sulphate-reducing bacteria. In addition to these processes, nanoparticles are generated around the cell surface as a result of heterogeneous nucleation on the cell surface layers or biofilms [11,12,13]. Examples of microbially induced precipitates include carbonate, silica and sulphate [14,15,16,17]. Microbes and nanoparticles, therefore, are closely associated in the natural environment.

Particles in the nanometre size range exhibit unique physical, chemical and catalytic properties that are different from those of bulk materials [18,19]. As a result, nanoparticles are used in a wide range of applications, such as catalysis, metal binding, and contaminant uptake [20,21]. For example, manufactured nanoparticles are known to be playing an increasing role in water, soil and air treatments; efficient energy production and storage; and medicine [4,22]. To facilitate these novel applications, nanoparticles are being synthesized both abiotically and biotically. Undoubtedly, large doses of synthetic nanoparticles are released into natural environments. However, little is known about the effects of nanomaterials on microbes in natural environments [23,24]. Therefore, it is necessary to study the interactions between microbes and nanoparticles in natural environments.

Another reason for elucidating microbe–nanoparticle interactions lies in the need to develop novel agents to eliminate pathogenic nanoparticles. The effects of nanoparticle toxicity on microorganisms are currently studied with the development of new bacterial strains that are resistant to current antibiotics [25,26]. The antibacterial effects of many kinds of nanoparticles, such as metals [27,28,29], metal oxides [30], fullerene [31,32,33] and nanoparticles coated with different functional groups [34,35,36], have been studied. These nanoparticles (metal, metal oxides and fullerene) were shown to be effective bactericides by affecting bacterial signal transduction [37,38], disrupting bacterial cell membranes and causing leakage [39,40,41], or depleting intracellular ATP levels [42,43,44].

The objective of this study was to simulate the potential toxicity and influences of nanoparticles on microbes in environments by assessing the interactions between nanoparticles and microbes in vitro. As nanoparticles are widely distributed in soils and sediments, gold nanoparticles, ludox and laponite can serve as model metal nanoparticles, colloidal silica and clay nanoparticles, respectively. To date, although there are several studies to investigate the interaction between nanoparticles and bacteria, the toxicity mechanism of nanoparticles on microorganisms needs further investigation [45,46,47,48]. In this work, we investigated nanoparticle–microbial cell interaction mechanisms. In particular, we studied the effect of nanoparticles on the iron-reduction capacity of *Shewanella putrefaciens* and the effect of microbes on nanoparticle aggregation.

## 2. Materials and Methods

### 2.1. Materials

An *Escherichia coli* strain DH5 (*E. coli*) and the iron-reducing bacterium *Shewanella putrefaciens* strain CN32 (*S. putrefaciens*) interacted with various nanoparticles. Luria–Bertani (LB) and tryptic soy broth (TSB) (Difco Laboratories, Plymouth, America) media were used for growing and maintaining *E. coli* and *S. putrefaciens*, respectively. The formula of LB medium is as follows: tryptone, 10 g/L; yeast extract, 5 g/L; sodium chloride (NaCl), 10 g/L. Sodium hydroxide (NaOH) was used to adjust the pH of the medium to 7.4 (this pH is suitable for the growth of *E. coli* strains), after sterilization by autoclaving at 121 °C. The formula of TSB medium is as follows: tryptone, 15 g/L; soy peptone, 5 g/L; and sodium chloride (NaCl), 5 g/L. Sodium hydroxide (NaOH) was used to adjust the pH of the medium to 7.2 ± 0.2, after sterilization by autoclaving at 121 °C. Gold nanoparticles are widely studied as model metal nanoparticles due their electrical, mechanical, thermal, chemical and optical properties [49,50]. Ludox is widely recognized as an excellent model for colloidal silica because ludox colloidal silica products are aqueous dispersions of silica particles in the low nanometre-size range that typically exhibit a narrow particle size distribution. Process design and control ensure quality and consistency [51,52]. Additionally, laponite was used as representative clay mineral nanoparticles because it can strongly interact with many types of chemical entities (from small molecules or ions, and natural or synthetic polymers to different inorganic nanoparticles) and can also be easily functionalized and readily degraded in a physiological environment, giving rise to non-toxic and even bioactive products [53,54]. Table 1 lists certain characteristics of the nanoparticles used in this study. Gold nanoparticles were synthesized following a previously published method [55]. The details of the synthesis process of gold nanoparticles were as follows: 300 mL of double-distilled deionized (DDDI) water and approximately 0.1 g of hydrogen tetrachloroaurate (AuCl_4_H) were added to a 500 mL flask. The hydrogen tetrachloroaurate solution was refluxed for 1 h under moderate magnetic stirring. Meanwhile, in a separate 50 mL beaker, 0.3 g of sodium citrate was added to 20 mL of DDDI water and sonicated for 1 min. The syringe was then filled with the entire citrate solution and quickly injected into the vortex of the hydrogen tetrachloroaurate solution. The solution was refluxed for an additional 20 min and then cooled to room temperature. This stock solution was stored in the plastic container in the refrigerator until needed. A 15% gold nanoparticle solution was obtained by adding 15 mL of the gold nanoparticle solution to 85 mL of distilled water. The ludox and laponite nanoparticles were purchased from Sigma (St. Louis, MO, USA) and Southern Clay Products Inc., (Austin, TX, USA) respectively.

### 2.2. Bacterial Growth in the Presence of Nanoparticles

To examine the susceptibility of *E. coli* to different nanoparticles (gold, laponite and ludox), *E. coli* cells were grown in 20 mL of a LB broth medium supplemented with nanoparticles of several different concentrations (the concentration of gold nanoparticles ranged from low to high as follows: 31 μg/L, 62 μg/L, 78 μg/L and 93 μg/L; the concentration of laponite and ludox ranged from low to high rank: 1.1 g/L, 2.1 g/L, 4.2 g/L, 5.3 g/L and 6.3 g/L at 37 °C. After 24 h, the cell numbers were counted to examine any inhibitory effects of the nanoparticles on *E. coli* growth. A 1 mL cell-nanoparticle suspension was removed from the experimental tubes, diluted and plated on LB agar plates. The plates were incubated at 37 °C for 24 h or longer. The number of colony forming units (CFU) was visually counted. The counts from three independent experiments were averaged. To quantitatively evaluate and compare the inhibition effects of nanoparticles on *E. coli* growth, we fitted the data with the following equation (nanoparticle susceptibility constant): Ln(N/N_0_)/x = C, where N is bacterial colony forming units (CFUs) at a given nanoparticle concentration, N_0_ is CFUs without nanoparticles and x is the concentration of nanoparticles (g/L) [59]. A higher C value means that the bacteria are more sensitive to the nanoparticles.

### 2.3. Effects of Nanoparticles on Iron Reduction Capacity

An iron-reducing bacterium, *Shewanella putrefaciens* strain CN32 (*S. putrefaciens*), was routinely cultured aerobically in tryptic soy broth (TSB) (30 g/L) from a stock culture, which was kept at –80 °C. After harvesting in TSB until the mid to late log phases, *S. putrefaciens* cells were washed with anaerobic bicarbonate buffer and resuspended in the bicarbonate buffer.

Laponite was made into slurries in the bicarbonate buffer (2.5 g/L NaHCO_3_) at different concentrations (i.e., 0.5, 1 and 2.5 g/L). These slurries were added to pressure tubes with a capacity of 23 mL (Balch tubes), then purged with N_2_/CO_2_ gas mix (80:20), and finally sealed with thick butyl rubber stoppers and autoclaving sterilization. Fe(III) in sterilized ferrihydrite (HFO) or ferric citrate as the sole electron acceptor was added to the tubes at a final concentration of 10 mM. HFO was synthesized following the procedure of [60]. Filter-sterilized lactate was used as the sole electron donor (5 mM). *S. putrefaciens* cells (1 × 10^8^ final concentration) were added to the treatment tubes with a sterile and anaerobic syringe. The abiotic controls consisted of tubes that received the same amount of nanoparticle suspension in the bicarbonate buffer in place of *S. putrefaciens* cells. All experiments were incubated at 30 °C without shaking. As for the experimental process, we also referenced some other studies [61,62].

The extent of microbial Fe(III) reduction in HFO or ferric citrate was monitored by measuring Fe(II) production with a ferrozine assay [63]. At select time points (the duration of the cultivation experiment was about 25 days, and the sampling time points of the mineral suspension were at the 0.07th day, the 0.75th day, the 2nd day, the 3rd day, the 5th day, the 10th day and the 25th day), 0.5 mL of mineral suspension, sampled with a sterile syringe, was added to a plastic tube with 0.5 mL of 1 N HCl (Ultrex grade, Sigma). The cell-mineral suspension was allowed to stand in HCl for 24 h before performing the ferrozine assay [64,65]. This extraction is denoted as 0.5 N HCl-extracted Fe(II) and has been shown to be effective for extracting microbially produced Fe(II), including the adsorbed form and Fe(II) in biogenic solids but not that in highly crystalline magnetite [66]. Aqueous Fe(II) was measured after centrifugation for 10 min at 14,100× *g* and 20 °C and analysed using the ferrozine assay. Surface-adsorbed Fe(II) was defined as the difference between ferrozine-extractable and aqueous Fe(II) concentrations.

### 2.4. Scanning and Transmission Electron Microscopy (SEM and TEM)

The mineralogical changes were studied with SEM and TEM. The SEM samples were prepared as previously described [67,68]. TEM was used as a complementary technique to examine nanoparticle–microbe associations. Glutaraldehyde-fixed cell-mineral suspensions were encapsulated in agar, sequentially dehydrated with varying proportions of acetone, imbedded with Spurr resin and polymerized at 60 °C [67,68]. The polymerized blocks were sectioned into ultrathin slices with a thickness of ∼60 nm using a Reichert-Jung ultramicrotome. Thin sections were observed with a Zeiss 10C TEM at an accelerating voltage of 100 kV.

### 2.5. Particle Size and Zeta Potential Measurements

The zeta potential and particle size of the laponite nanoparticles were measured with a commercial Zetasizer (Nano ZS, Malvern, PA, USA) using bicarbonate buffer as an electrolyte. Five measurements were conducted for each sample, and the averages were reported.

## 3. Results

### 3.1. Inhibitory Effects of Nanoparticles on Microbial Growth

Four types of nanoparticles at different concentrations were tested with *E. coli* in LB medium to observe the effects of the nanoparticles on bacterial growth. In general, the number of bacterial colonies grown on LB plates decreased with increasing nanoparticle concentrations (Figure 1), suggesting that the tested nanoparticles inhibited microbial growth but to different extents (Figure 1). The presence of gold nanoparticles at a concentration of 31 μg L^−1^ decreased bacterial growth to 18.5% of that without nanoparticles. The number of bacterial colonies in the presence of more than 62 μg L^−1^ gold nanoparticles was reduced to 1.59% of that in the absence of gold nanoparticles (Figure 1a). Ludox and laponite nanoparticles exhibited similar inhibitory effects on the growth of *E. coli.* but to different extents (Figure 1b). Ludox 7 and ludox 12 have the same chemical composition but different particle sizes. Notably, the 12 nm ludox nanoparticles had a stronger inhibitory effect on the growth of *E. coli* under low nanoparticle concentrations (1.1 g/L, 2.1 g/L) (Figure 1b). The C value of *E. coli* to gold nanoparticles was 7.20 × 10^4^ L/g, much higher that of other nanoparticles (C_Ludox 7 nm_ = 1.59 L/g; C_Ludox 12 nm_ = 8.73 L/g; C_Laponite_ = 1.57 L/g), suggesting that *E. coli* is more sensitive to gold nanoparticles than ludox or laponite (Figure 1c,d).

*E. coli*, a Gram-negative bacterium, has only a thin layer of peptidoglycan and a more complex cell wall with two cell membranes (an outer membrane and a plasma membrane). The outer membrane of Gram-negative bacterial cells controls the permeability of many molecules in and out of cells. For this reason, under certain conditions, Gram-negative bacteria are generally more resistant to many chemical agents than Gram-positive cells [69]. To further understand *E. coli*’s response to nanoparticle stress, bacterial morphology and nanoparticle–cell associations were examined with SEM. No aggregation was observed in the absence of nanoparticles (Figure 2a), but *E. coli* cells were aggregated after the addition of gold nanoparticles (Figure 2b). *E. coli* cells were aggregated at a gold nanoparticle concentration as low as 62 μg/L (Figure 2b). Laponite nanoparticles also caused the aggregation of *E. coli* (Figure 2d). Furthermore, nanoparticles themselves also aggregated (Figure 2b,d).

TEM was used as a complementary technique to examine thin sections showing nanoparticle–cell associations. Nanoparticles appeared to disrupt the cell walls of *E. coli* cells (Figure 3). TEM micrographs of *E. coli* treated with gold nanoparticles showed that the gold nanoparticles aggregated around cell debris and “unhealthy or dead” cells but not around “healthy” cells (Figure 3a). An *E. coli* cell coated with gold nanoparticles appeared to have lost its cell membrane (Figure 3a). Gold nanoparticle clusters were found to anchor to the bacterial cell wall (Figure 3b), possibly at sites that were rich in negatively charged functional groups. Some nanoparticles were observed to penetrate the cell membrane (Figure 3b), possibly causing cell death and leaking of the internal contents of the cell. Cell lysis would account for the decreased bacterial growth observed in the presence of nanoparticles (Figure 1). Thus, the collective effects of the nanoparticles anchoring onto the cell surface and penetrating into the cell wall and cell plasma may have eventually caused the complete annihilation of bacterial cells. The penetrating gold nanoparticles appeared to have different shapes than the original spheres.

TEM micrographs of *E. coli* treated with laponite nanoparticles showed similar phenomena (Figure 3c,d). The laponite nanoparticles appeared to be clustered around the cell surface, even though both the cell wall and laponite nanoparticles had negative surface charges. The laponite nanoparticles were also able to penetrate the *E. coli* cell wall (Figure 3d).

### 3.2. Bioreduction of Fe(III) with Laponite

The current hypothesis is that the surface-attached nanoparticles around cells (Figure 3) might hinder the electron transfer process. Solid Fe (III) (HFO) and aqueous Fe (III) (ferric citrate) bioreduction experiments were used to test this hypothesis. *S. putrefaciens* was able to reduce the solid Fe (III) in HFO and aqueous Fe (III) when laponite nanoparticles were absent (Figure 4; 1 × 10^8^ cell mL^−1^). Structural Fe (III) bioreduction occurred most rapidly and to the greatest extent in the absence of laponite nanoparticles. However, in the presence of laponite nanoparticles, structural Fe (III) was not effectively reduced even at a low laponite concentration (0.5 g L^−1^) (Figure 4a). In comparison with structural Fe (II), aqueous Fe (III) was reduced relatively slowly from the solution in the absence of laponite (Figure 4b). In contrast to the inhibitory effect of the laponite nanoparticles on solid Fe (III) bioreduction (Figure 4a), the addition of laponite increased the rate of aqueous Fe (III) bioreduction in the early days of incubation (0–2 days), suggesting that laponite nanoparticles facilitated the bioreduction of aqueous Fe (III) (Figure 4b). The amount of Fe (II) sorbed onto the laponite nanoparticles was measured in terms of bioreduced ferric citrate, and the data indicated that the laponite nanoparticles strongly scavenged biogenic Fe (II) (Figure 4c).

### 3.3. Zeta Potential and Particle Size

After the cessation of HFO or ferric citrate bioreduction, the zeta potential of the laponite nanoparticles was measured and showing differences from that of abiotic controls (Figure 4d). Over the entire range of concentrations, the zeta potential values of the laponite nanoparticles were strongly negative. The zeta potential of all the bioreduced samples was higher (less negative) than that of their corresponding abiotic controls. Because the zeta potential is a function of particle surface charge, adsorbed ions, and the surrounding medium in which the particle is suspended [70], the higher zeta potential for the bioreduced samples was probably due to surface-absorbed Fe (II). Zeta potential reflects the effective charge on the particles and is therefore related to the electrostatic repulsion between them. The increase in zeta potential for the bioreduced samples (less negative) might be a reason for the aggregation of laponite nanoparticles (Figure 5a). *S. putrefaciens* cells also aggregated as a result of the addition of laponite nanoparticles (Figure 5b). Laponite nanoparticles were encrusted by cell extracellular polymeric substances (EPS) (Figure 5). The formation of large amounts of cell/nanoparticle aggregates resulted in an increase in the turbidity of the treatment tubes. The laponite nanoparticle size increased after the bioreduction of the Fe (III) in ferric citrate (Table 2), but it was not possible to make such a comparison for the HFO treatments because different particles were present in the abiotic control (HFO and laponite) and bioreduced sample (*S. putrefaciens* cells, HFO and laponite). A higher laponite nanoparticle concentration promoted *S. putrefaciens* cell and laponite nanoparticle aggregation for both the ferric citrate and HFO systems.

## 4. Discussion

### 4.1. Inhibitory Effects of Nanoparticles on Bacterial Growth

The attachment of nanoparticles to bacteria depends on several factors, such as the particle composition and shape, and the cell and particle surface characteristics, which determine the sign and magnitude of the interaction forces between electrostatic attractions [71,72,73,74]. Because bacteria are negatively charged (*E. coli*) and gold nanoparticles are positively charged under the experimental pH (7), the interaction force between them would be attractive. The attractive force would explain the great inhibitory effect of gold nanoparticles on *E. coli* growth (Figure 1). Previous studies showed that the toxicity of positively charged nanoparticles with negatively charged bacterial membranes was more profound than that of neutral and negatively charged nanoparticles [34,73]. Because the laponite nanoparticles are negatively charged and the interaction would be repulsive, the inhibitory effect was less significant than that of gold nanoparticles (Figure 1). However, despite the negative surface charge of the laponite particles (Table 1), these nanoparticles still interacted with the cells, causing cell damage and ultimately death [47,75]. Our data not only supported the surface charge-associated effects but also showed that negatively charged nanoparticles somehow caused bacterial death (Figure 1 and Figure 3).

Once attached to the cell surface, nanoparticles disrupted the bacterial surface, causing membrane permeability and finally cell death by binding to and penetrating the cell membrane (Figure 3). Our data are consistent with previous results and clearly show that nanoparticles not only coat the cell surface but also penetrate into bacterial cells [25,29]. As a result of these combined effects, membrane integrity was adversely affected (Figure 3), and cell lysis may have occurred. These adverse effects would account for the decreased growth rate (Figure 1) and decreased bioreduction (Figure 4a). A similar effect was reported by several studies, suggesting that the changes in the membrane morphology caused by nanoparticles may produce a significant increase in membrane permeability and cause the cells to be incapable of properly regulating transport through the plasma membrane [29,76,77].

Our data showed that the morphology (from granulated or acicular to coatings) of the nanoparticles changed after they entered bacterial cells (Figure 3), which was especially true for gold nanoparticles. Konishi and colleagues reported [78] that *Shewanella algae* is capable of reducing Au from +7 (in the form of AuCl_4_^−^ ions) to elemental gold, depositing nanoparticles of gold in the periplasmic space of the cells. It was unlikely that a similar mechanism was operating in our experiments because elemental gold nanoparticles were used. However, a dissolution–reprecipitation mechanism could not be excluded at present because organic materials from lysed cells could possibly solubilize gold nanoparticles, which would facilitate the uptake of ionic gold into cells [79,80].

### 4.2. Physically Blocking Electron Transfer from Microbial Cells to Electron Acceptors

Our data showed that laponite nanoparticles strongly inhibited solid-phase Fe (III) bioreduction but not aqueous Fe (III) bioreduction. This differential effect may be explained by different mechanisms of electron transfer between these two types of bioreduction. The biological reduction of solid Fe (III) is mediated by electron transfer proteins that are localized in the cytoplasm [81,82]. The electron transfer process can be facilitated by electron shuttling compounds, such as anthraquinone-2,6-disulfonate (AQDS) [83]. In the absence of electron shuttles, direct contact is the main mechanism for solid Fe (III) bioreduction [84]. This direct contact may be interrupted if other solids are present in the system, such as nanoparticles. Microbial cells might be coated by nanoparticles, and this coating separated HFO from bacterial cells, thus inhibiting the electron transfer efficiency. In addition, negatively charged nanoparticles could coat positively charged HFO solids, thus further inhibiting the electron transfer process. Our evidence of increased particle aggregation would certainly support this mechanism (Figure 5 and Table 2).

Our data (Figure 4b) showed that laponite particles had an insignificant effect on the extent of aqueous Fe (III) bioreduction but slightly enhanced the rate of bioreduction. This may be explained by the fact that aqueous Fe (III) could readily diffuse through nanoparticles and come into contact with the bacterial surface for electron transfer [85]. The movement and diffusion of solid nutrients in the medium were more restricted than those of liquid nutrients. The slightly increased rate of bioreduction by laponite nanoparticles may be due to the removal of inhibitory Fe (II) from the cell surface by its sorption onto the laponite nanoparticle surface. Likewise, Fe (III) may also be similarly adsorbed by laponite nanoparticles. However, due to the high abundance of aqueous Fe (III) in ferric citrate solution, this adsorption might not affect the rate of bioreduction.

Laponite nanoparticles have a high sorption capacity for heavy metals and humic acids [54,58]. Biogenic Fe (II) is toxic to bacteria, and its sorption to the bacterial surface could inhibit further bioreduction of iron oxides [65] and clay minerals [86]. Increased laponite concentrations would sorb more Fe (II) and promote bioreduction. As shown in Figure 4b, given sufficient incubation time and a cell concentration that is not rate-limiting, the ultimate extent of bioreduction should converge regardless of the laponite particle concentration. Because clay minerals are common components in soils and colloids in aquatic environments and act as major sinks for heavy metals [87,88], they may play an important role in not only influencing microbial bioreduction and growth but also affecting the distribution, mobility and bioavailability of metals [58].

### 4.3. Effects of Bacteria on the Aggregation of Nanoparticles

Nanoparticles can disrupt the bacterial cell membrane (Azam et al., 2020) and affect central metabolism [37,42]. On the other hand, bacteria may respond to nanoparticle stress. For example, bacteria can cause silver or ZnO nanoparticle aggregation, resulting in the loss of nanoparticle antibacterial activity [39,42,71].

Our SEM images showed macroscopic aggregates composed of nanosized particles, cells and EPS (Figure 2, Figure 3 and Figure 5). The presence of cells and their products significantly enhanced nanoparticle sizes (Table 2). Among biological agents, EPS appear to play a particularly important role in particle aggregation [89,90]. It is very likely that the aggregation of cells/nanoparticles protects the cells from exposure to environmental stresses, including nanoparticles. It has been reported previously that the aggregation of *S. oneidensis* cells was associated with oxidative stress, including that imposed by antibiotics [91]. It is possible that the aggregation of cells/nanoparticles decreases the contact area between cells and nanoparticles and decreases the bioavailability of nanoparticles in solution. Szomolay et al. (2005) reported that EPS promotes cell aggregation [92]. It is also possible that cell/nanoparticle aggregation was mediated by EPS because EPS polymers may attach to particles and bridge them to form aggregates, even though the particles are electrostatically repulsive [93].

The difference in the particle diameters clearly shows the effect of bacteria on nanoparticle aggregations. Thus, possibly as a passive defence mechanism, bacteria may detoxify nanoparticles by promoting their aggregation with EPSs. We attribute this phenomenon to the difference in the high zeta potential and microbially produced EPS.

## 5. Conclusions

The effects of the addition of nanoparticles on the behaviour of bacteria are twofold. First, the inhibition of bacterial growth was observed, as evidenced by lower CFU numbers. Electron microscopy proved that the bactericidal role of nanoparticles is to cause damage to the membrane permeability and structure. Moreover, nanoparticles block electron transfer from the cell membrane to solid electron acceptors by physically obstructing direct contact between cells and electron acceptors. Nanoparticles, strong sorbents of heavy metals, may be beneficial to microbes in aquatic environments. On the other hand, microbes might cause nanoparticle aggregations.

## Figures and Tables

**Figure 1 microorganisms-11-00542-f001:**
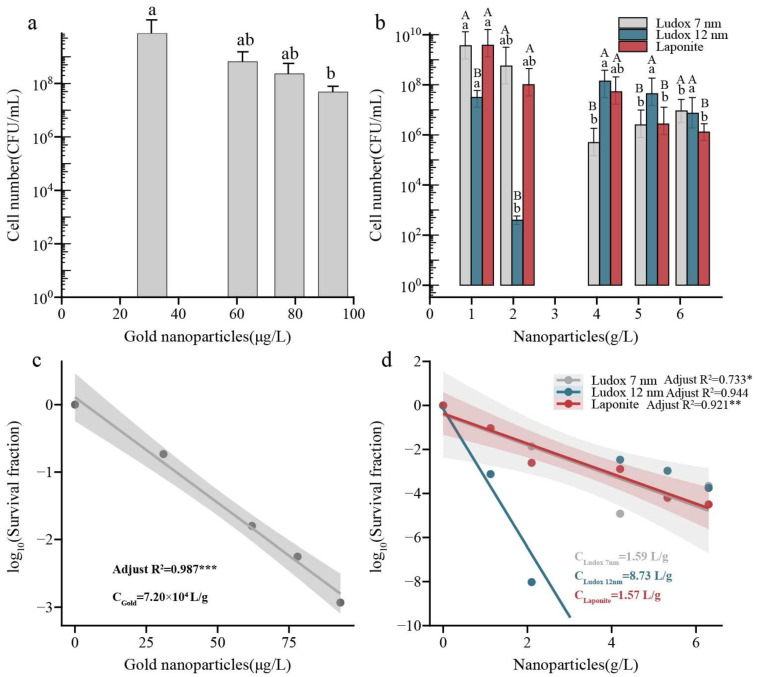
Number of *E. coli* colonies from LB agar medium containing nanoparticles at different concentrations. The initial cell concentration (CFU/mL) was 10^10^. (**a**) Gold nanoparticle; (**b**) ludox 7 nm, ludox 12 nm, and laponite; (**c**) effect of gold nanoparticles on the bacterial susceptibility constant; (**d**) effect of ludox 7 nm, ludox 12 nm, and laponite nanoparticles on the bacterial susceptibility constant. Differences in cell concentration (CFU/mL) among nanoparticles at different concentrations were obtained by one-way ANOVA (*p* < 0.05) followed by Turkey’s multiple comparison tests. Lowercase letters represent the difference in the number of bacteria at different concentrations of the same type of nanoparticles, and uppercase letters represent the difference in the number of bacteria at the same concentration of different types of nanoparticles. *, 0.01 < *p* < 0.05; **, 0.001 < *p* < 0.01; ***, *p* < 0.001.

**Figure 2 microorganisms-11-00542-f002:**
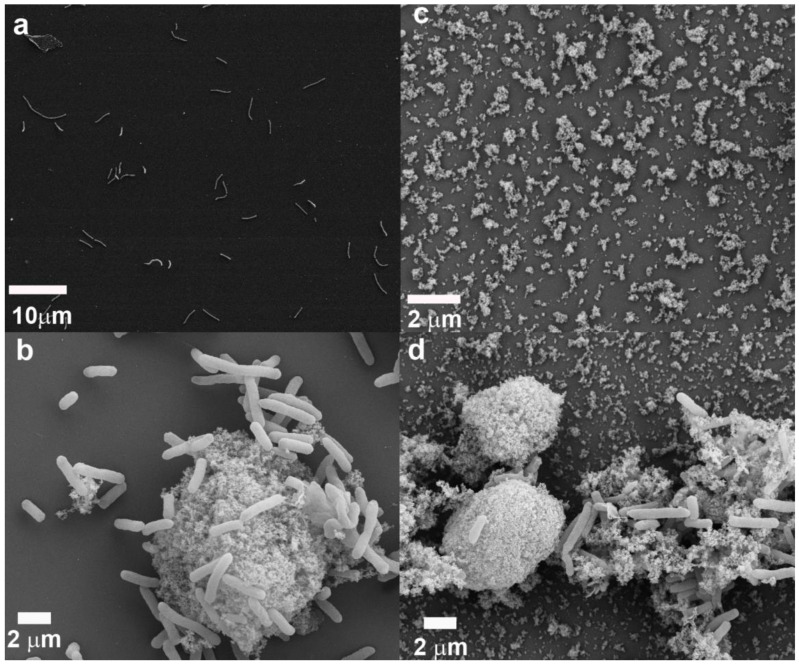
SEM images of *E. coli* cells and nanoparticles: (**a**) *E. coli* cells, (**b**) gold nanoparticles with *E. coli* showing the aggregation of nanoparticles and bacterial cells, (**c**) laponite nanoparticles, and (**d**) laponite nanoparticles with *E. coli*.

**Figure 3 microorganisms-11-00542-f003:**
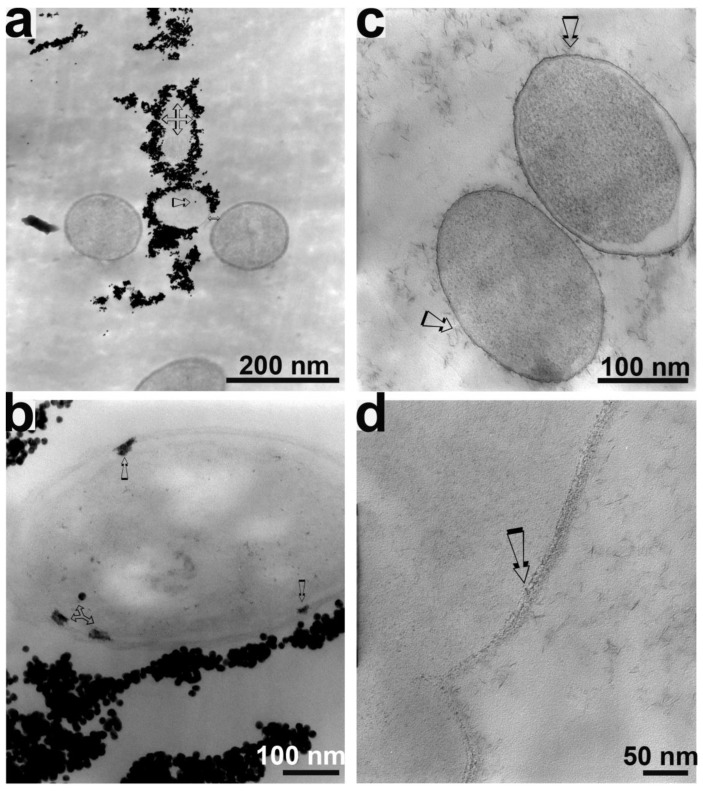
TEM images of nanoparticles and *E. coli* showing the interaction of the bacteria with the nanoparticles. (**a**,**b**) Gold nanoparticles with *E. coli*, (**c**,**d**) laponite nanoparticles with *E. coli*. Arrows in the figure represent the cell membrane of *E. coli* cells and damage of nanoparticles to cell membrane of *E. coli* cells. Arrow represent different morphology of nanoparticles.

**Figure 4 microorganisms-11-00542-f004:**
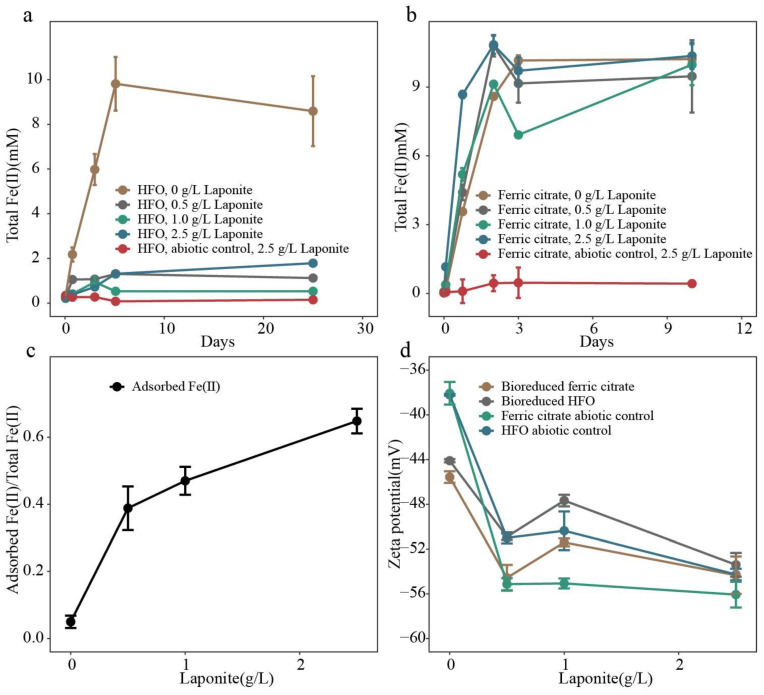
Microbial reduction of Fe (III) in HFO and ferric citrate by *S. putrefaciens*. (**a**) Change in 0.5 M HCl-extractable Fe (II) with time as *S. putrefaciens* reduced Fe (III) in HFO. Fe (III) (10 mM) in HFO was used as the sole electron acceptor, and lactate (5 mM each) was used as the sole electron donor. (**b**) Change in 0.5 M HCl-extractable Fe (II) with time as *S. putrefaciens* reduced Fe (III) in ferric citrate. Fe (III) (10 mM) in ferric citrate was used as the sole electron acceptor, and lactate (5 mM each) was used as the electron donor. (**c**) Surface adsorption of biogenic Fe (II) (from ferric citrate) as a function of laponite concentration in bicarbonate medium. (**d**) Zeta potential of laponite nanoparticles after bioreduction.

**Figure 5 microorganisms-11-00542-f005:**
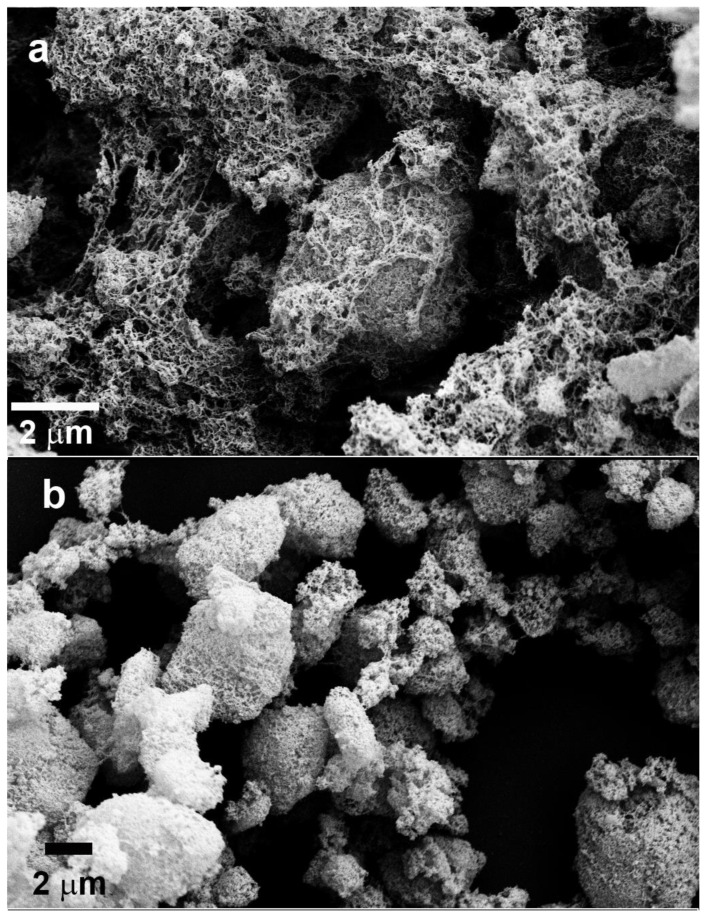
SEM image of *S. putrefaciens* with bioreduced HFO or ferric citrate in the presence of laponite nanoparticles showing biofilm and aggregated nanoparticles. (**a**) HFO with laponite and *S. putrefaciens* cells, (**b**) ferric citrate with laponite and *S. putrefaciens* cells. HFO, ferrihydrite.

**Table 1 microorganisms-11-00542-t001:** Summary of nanoparticle properties.

Sample	Composition	Particle Size	BET Surface Area (m^2^/g)	CEC (Mequiv/100 g)	pH_PZNPC_	Surface Site Density (Sites/nm^2^) = mol/g	Shape
Gold	Au	8	na	na	2.5–3.5 ^a^	na	sphere
Ludox 7 ^b^	SiO_2_	7	220	na	3.5–3.9	na	sphere
Ludox 12 ^b^	SiO_2_	12	215	na	3.5–3.9	na	sphere
Laponite ^c^	Na_0.8_[Mg_5.4_Li_0.4_]Si_8_O_20_(OH)_4_	20	300	48.1	9.5	3.2	plate

^a:^ [56]. ^b^: [57]. ^c^: [58]. na, null value.

**Table 2 microorganisms-11-00542-t002:** Particle size distribution of laponite nanoparticles after the bioreduction of the Fe (III) in HFO or ferric citrate.

Laponite Concentration (g/L)	FC		F		HC		H	
	Diameter (nm)	Percentage (%)	Diameter (nm)	Percentage (%)	Diameter (nm)	Percentage (%)	Diameter (nm)	Percentage (%)
0	~	~	654 ± 72.3	100	635 ± 88.3	100	864 ± 126	100
0.5	142 ± 55.9	58.5	72.6 ± 6.03	12.3	458 ± 19.3	100	160 ± 26.6	27
	592 ± 196	41.5	593 ± 75.9	87.7	~	~	883 ± 180	73
1	99.9 ± 19.6	28.6	104 ± 21.3	53.3	615 ± 14.3	100	146 ± 25.9	25.4
	453 ± 102	71.4	550 ± 90.3	46.7	~	~	747 ± 148	74.6
2.5	123 ± 37.5	38.8	160 ± 71.5	75	80.9 ± 6.98	5.3	174 ± 40.6	26.1
	518 ± 170	61.2	815 ± 297	18.3	934 ± 111	94.7	976 ± 240	73.9
	~	~	5000 ± 613	6.7	~	~	~	~

Note: F represents the Fe(III) in ferric citrate; H represents the Fe(III) in HFO; 0–2.5 indicates the laponite concentration in g/L; C indicates the abiotic control. HFO, ferrihydrite.

## Data Availability

The data presented in this study are available on request from the corresponding author. The data are not publicly available due to privacy or ethical restrictions.

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
