# Peer review of "Effects of Different Nanoparticles on Microbes"

_microorganisms, 2023, doi:10.3390/microorganisms11030542_

Round 1
Reviewer 1 Report
Dear editor,
In the manuscript "Effects of Different Soil Nanoparticles on Microbes", the authors used a combination of techniques to investigated the interaction of nanoparticle and two strains of bacteria. They found that nanoparticle can interact with cell surface, block access of nutrients and cause cell death. The work is generally well conducted, but further improvement may be required in the interpretation of results. In addition, the title refers to soil nanoparticle, but in manuscript only Ludox was tested. Therefore, the title may be changed to be more specific.
Specific comment:
Line 8: the first sentence provides little information, therefore is suggested to rephrasing it
Line 23-24: line 20 stated Fe(III) reduction in aqueous form was accelerated, contradicting the ‘not have any significant effect‘
Line 24-25: Is Fe (II) considered toxic heavy metal? There was no mention of scavenging of Fe (II) in early sentences, therefore it is unclear how this conclusion was made
Line 27: there lacks a final implication statement.
Introduction
Line 62-65: what is already known regarding the environmental impact of nanoparticles released from industrial activities? The author only stated what needs to be done, therefore is inappropriate for introduction purposes.
Line 82-83: the author stressed that the object was to assess the toxicity in the environment, but the study was in vitro, therefore cannot achieve the object.
Line 84: what is the “they” refer to?
Line 85-86: this is an overstatement, there are several/many studies investigated the interaction of nanoparticle and bacteria, such as Antimicrobial activity of iron oxide nanoparticle upon modulation of nanoparticle-bacteria interface; and Hydrophilic nanoparticles that kill bacteria while sparing mammalian cells reveal the antibiotic role of nanostructures.
Materials and methods
Line 93: LB and TSB medium, please specify their components, the concentration of each component, the supplier information is incomplete. Also, most of the reagent information is incomplete across the method section.
Section 2.2: key information is missing here. What is the start cell number, and how do ensure each treatment (replicate) has the same starting cell number; how the incubation experiment was performed with or without shaking?; “several different concentrations” what are these concentrations?
Line 120: “in the buffer” in which buffer?
Line 122-123: please rephrase, the sentence is unclear
Line 124: “microwave sterilized” Is this a valid sterilization technique?
Line 128: how this is ensured that the cell concentration was 1E8?
Line 130: was there any anaerobic condition indicator added to ensure no gas leakage?
Line 135-136: during incubation, could it be possible that Fe(II) re-oxidized into Fe (III) by oxygen in the air?
Results:
Figure 1a: What is the colony number in the control experiment?
Line 180-184: these are methods
Figure 1B: some of the changes seems non-linear, such as at 7nm and 12 nm Ludox. Could the author provide any explanation.
Line 185: the ludox12 seems to have a greater slop compared with the gold nanoparticle, therefore contradicting this statement
Line 190-192: these are more like discussion
Figure 2c: where is the e. coli cell? The text stated laponite caused E. coli aggregation (line 198).
Line 214-215: This text seems to have a different font size from the rest of the manuscript.
Figure 3: what do the arrows show?
Line 225: which figure shows this?
Line 229-230: Should this be “when laponite nanoparticles were absent?”, what is the 1*108 referred to?
Line 234-236: this sentence is unclear, please rephrase
Line 239-240: Is this reflected in the figure? (4b). It seems the acceleration was only observed at 2.5 g/L, but not at any other laponite concentration.
Line 241-243: where the Fe(II) only quantified on laponite? The method didn’t mention this. which results are used to support this statement?
Line 254-256: grammar issue
Line 258-261: more like a discussion
Figure 4c is not mentioned in the text
Line 264: which panel of fig 5?
Line 271-272: should there also be HFO in the bioreduced sample?
Figure 5: what is the information of this figure? The two panel looks very similar. Also needs to define HFO for figure legend.
Discussion
Line 287: but is this contradictory to the acceleratory effect of ludox12?
Line 294: Please add appropriate citation(s) to support this.
Line 299: what are these previous results? Please add citation
Line 302-303: were these “adverse effects” also observed in the Fe reduction experiment, as adding nanoparticles to Fe citrate accelerated the reaction rate
Line 327: Figure 3 is the case of e.coli, but not S. putrefaciens CN32
Table 2: there are several issues with this table. The sample name column is confusing, could use the real concentration instead; what do the diameter and percentage sample mean? Was there any significant difference between control and treatment? What does ~ stand for, and why there are two values for each cell?
Line 338-339: reference
Line 343-344: please elaborate on this further.
Line 350-351: this sentence is unclear
Line 367-369: is the antibiotic effect considered oxidative stress?
Line 387-388: this may need further evidence
Reviewer 2 Report
The study of the effect of nanoparticles on the growth of microorganisms is an important area that reveals patterns of physicochemical interactions between bacterial cells and nanoparticles. The authors did a great job, but in my opinion, the methods section should be described in more detail and the results should be clearly explained.
Lines 85-86: This statement is controversial. For example, the mechanisms of action of gold, silver, and laponite nanoparticles are described in the following works:
https://doi.org/10.1021/acsami.9b03527
https://doi.org/10.1021/acsnano.7b02035
https://doi.org/10.1007/s00775-019-01717-7
https://doi.org/10.3390/ma15051799
Lines 92-94: Hereinafter, the generally accepted writing of microorganisms should be used: at the first mention - the full, and then the abbreviated name of the genus:
“Escherichia coli strain DH5 and the iron-reducing bacterium Shewanella putrefaciens strain CN32…” and then - "... maintaining E. coli and S. putrefaciens, ...".
Lines 99-100: The conditions for the synthesis of gold nanoparticles should be described in more detail. It is also desirable to give the characteristics of nanoparticles, their size range and spectra confirming the synthesis of gold nanoparticles.
Line 110: Write the values of all concentrations of different types of nanoparticles that affect microorganisms.
Line 116: A method for studying the iron-reducing ability of bacteria in the presence of laponite is not entirely clear. There are many references to methods described in other sources, which makes it difficult to understand an experiment design.
Line 119: How were the growth phases of Shewanella putrefaciens determined, and how long was the incubation period?
Line 123: Balsch tubes - typo, replace with Balch tubes.
Line 124: Which mode was used for microwave sterilization?
Line 133: What was the duration of the experiment and sampling time?
Lines 163-165: Bacteria reduction with 62 μg/L gold nanoparticles was greater than 31 μg/L. Perhaps there is a typo and the concentrations of 18.5% and 1.59% are mixed up? What concentration of nanoparticles reduces the number of bacteria by 18.5%? Or is it the average of three high concentrations?
Line 296: In my opinion, this statement does not clarify the mechanism of interaction between nanoparticles and cells. Even if the adsorption of nanoparticles on the surface was shown, this does not confirm the association with cell death. How can you explain the absence of nanoparticles on the surface of living cells (Fig. 3A)?
Line 308: Specify how the morphology of nanoparticles has changed.
Reviewer 3 Report
Effects of Different Soil Nanoparticles on Microbes
This study explored the detrimental effects of nanoparticles on microbes. The current study investigated the interactions between microbes and nanoparticles by exploring the inhibition effects of gold, ludox and laponite nanoparticles on Escherichia coli Luria–Bertani (LB) medium at different nanoparticle concentrations. This work is interesting and can help us better understand the cell membrane disruption due to nanoparticle accumulation on cell surfaces, resulting in cell death. Overall, this is an outstanding study with excellent data, prudent statistical analysis and interpretation. The paper is generally well-written but the methodology framework is unclear and lacks potential references. However, I propose some suggestions that can improve the document. My overall recommendation as a reviewer to the editor is that the manuscript can be considered for publication after major revision.
Introduction:
The introductory section of this article lacks critical and in-depth analysis of the available literature and comprehensive relevant work.
Comment 1: As nanoparticles is a global issue, therefore, authors should incorporate quantitative data about the concentration of reported nanoparticles of different countries and it would be good to cite and relate some other studies that investigated similar aspects.
Comment 2: I suggest you add 2-3 introductory sentences about gold, ludox and laponite. There is no information given about how these concentrations were selected.
Comment 3: Explain the novelty of the work in a paragraph and make a comparison with the literature.
Comment 4: Besides the wide use of nanoparticles, how about its residual level and existing characters in the natural environment? The reason to select gold, ludox and laponite as a model nanoparticles should be explained.
Comment 5: There are several citations in methods that should be replaced with primary references.
Comment 6: How nanoparticles inhibited microbial? Discuss the detailed mechanism.
Comment 7: “Our data not only supported the surface charge-associated effects but also showed that negatively charged nanoparticles somehow caused bacterial death” needs more explanation. Authors should precisely focus on and strengthen the discussion section with published research work.
Comment 8: I suggest you conduct a multivariate statistical analysis and determine the interaction effect among key parameters of this study.
Round 2
Reviewer 2 Report
Thanks to the authors for a great job. I also thank you for clarifying some aspects of the work that I did not understand.